# Mineral Composition and Graphitization Structure Characteristics of Contact Thermally Altered Coal

**DOI:** 10.3390/molecules27123810

**Published:** 2022-06-14

**Authors:** Huogen Luo, Wenxu Liang, Chao Wei, Dun Wu, Xia Gao, Guangqing Hu

**Affiliations:** 1College of Safety Science and Engineering, Anhui University of Science and Technology, Huainan 232000, China; 15856577135@163.com; 2State Key Lab of Coal Mine Safety Technology, Shenyang Research Institute, China Coal Technology and Engineering Group, Fushun 113122, China; wenxu_liang@126.com; 3Exploration Research Institute, Anhui Provincial Bureau of Coal Geology, Hefei 230088, China; 2017170534@mail.hfut.eud.cn; 4Key Laboratory of Intelligent Underground Detection Technology, School of Civil Engineering, Anhui Jianzhu University, Hefei 230601, China; 5School of Architecture & Urban Planning, Anhui Jianzhu University, Hefei 230601, China; gaoxia@ahjzu.edu.cn

**Keywords:** mineral composition, graphitization, structure, contact thermally altered

## Abstract

Contact metamorphism in coal is usually characterized by a rapid, brief, and exotherm reaction that can change the geothermal gradient. In this process, coal adjacent to the intrusive body can form thermally altered coal-based graphite (TACG). In order to further study the structural changes of TACG at different distances from the intrusive body, four TACG samples were collected in the Zhuji coal mine in the Huainan Coalfield, North China, and their vitrinite reflectance and Raman spectra were measured using polarizing microscopy and Raman spectroscopy. The results showed that: (1) affected by the temperature and stress of magmatic hydrothermal intrusion, the clay minerals in the coal seams appeared distributed in strips; the occurrence of ankerite and pyrite in the coal seams near the magmatic intrusions could be due to a late magmatic hydrothermal mineralization; (2) the R_max_ − R_min_ correlation for the TACG samples under study showed that thermal metamorphism was the main factor leading to the graphitization of the TACG samples, without an obvious pressure effect; (3) with the increase of the graphitization process, the D- and G-band showed some similar changes, specifically, their peak positions shifted to lower wave numbers, and the full width at half maximum (FW_G_ and FW_D_) gradually decreased; the difference was that the intensity of the G-band increased, while that of the D-band decreased; (4) the graphitization degree of the TACG samples increased with the increase of the transverse size of the crystals, while the FW_G_ and FW_D_ values of the G- and D-band decreased; (5) in comparison to natural graphite, the TACG still presented structural defects.

## 1. Introduction

Carbonaceous materials (CM) have attracted research interest worldwide in recent years. In the process of geological evolution, CM in nature can be gradually transformed into disordered graphite. With the continuous improvement of the order degree, ordered graphite can form [1,2].

Thermal events related to igneous intrusion are relatively common in many countries in the world, and a large number of studies have been conducted on the impact of igneous intrusion on coal [3,4,5,6,7,8,9,10,11,12,13,14,15,16,17]. Especially, research has been carried out on some representative coalfields i.e., the Illinois, Raton and San Juan Basins in the USA, the Gunnedah Basin and Bowen Basin in Australia, and the Qinshui, Fushun and Tiefa Basins in China [6,8,10,11,18,19,20,21]. In these coalfields, some coal seams and their surrounding rocks were invaded by dikes, sills, and other types of igneous rocks. The heat generated by this invasion may be as high as 600 °C, or even higher during invasion, causing significant changes in the coal. More generally, the higher the temperature, the more obvious the changes are. Therefore, it is possible to trace the changes in the structure of coal at different distances from the intrusive body.

More generally, these effects of heat are local and variable and, geologically speaking, they occur over a comparatively short period. This kind of alteration contrasts with the normal progression of coal rank which is a regional and slow process in response to normal, rather than abnormal, upper crustal temperatures [22]. Compared with coal during normal coalification, thermally altered coal has typical physicochemical characteristics [10,11,13,19,20]. For example, the vitrinite reflectance of coal increases, but the moisture, volatile matter, and hydrogen content decrease as the distance from the intrusion decreases. Significantly, when coal is transformed into graphite by contact thermal metamorphism, the resulting graphite can be called thermally altered coal-based graphite (TACG) [23].

The technical methods commonly used in TACG research are usually two: optical microscopy (high-resolution electron microscopy (HREM)) and spectroscopy (X-ray diffraction (XRD) and Raman spectroscopy). HREM was used to study coal-based graphite series samples. The results showed that the transition from coal to graphite formed a two-dimensional graphite lattice (aromatic graphite) in the first stage, then three-dimensional lattice graphite microcrystals (microcolumn graphite and soft graphite), and finally flat graphite [24]. In the transformation from amorphous to crystalline, the basic process was the lateral expansion of the two-dimensional graphite carbon layer and the increase of the thickness of the graphite carbon layer. If the graphite carbon layer was regarded as a complete lattice region, then the random carbon atom region between the graphite carbon layers was the “Defect” region of the coal structure. This “Defect” region was called the original structural defect in the graphitization process of coal. Therefore, the whole graphitization process was the process of reducing the “Defect” region of the original structure until it was finally eliminated. Because of the advantage of causing minimum damage to CM chemical structure, XRD can effectively study the crystal structure characteristics of CM [24]. Five structural parameters, including amorphous carbon fraction (*A*), aromaticity (*f_a_*), interlayer spacing of the crystal structure (*d_002_*), and crystallite size (*L_a_* and *L_c_*), are commonly used to define the carbon stacking structure of CM [25]. The Raman spectra of CM are usually divided into first-order and second-order regions [26]. The first-order region, especially the graphite zone (G) and the defect zone (D) at 1582 and 1357 cm^−1^, respectively, reflects the structural order of the CM [27,28,29,30]. For perfect single graphite crystals, the G-band becomes visible at 1580 cm^−1^, accompanied by a strong C-C stretching mode (*E*_2g2_ vibration mode), which indicates the stretching vibration of *sp*^2^ bonds in hexagonal aromatic molecules. On the contrary, for disordered structural CM, the D-band (in-plane defect), which represents the graphite lattice vibration and has *A*_1_g symmetry, appears. The disorder degree of carbon structure in CM can be distinguished by comparing the D- and the G-band in Raman spectra [31].

For TACG, previous research mainly focused on the characterization of coal-based graphite, the integrity of coal crystals, and the ordering of coal structures. There is still a lack of in-depth research on whether the “Defect” region of TACG is caused by original structural defects or by secondary structural defects. Therefore, in this study, TACG samples with different distances from the intrusion were collected, and the vitrinite reflectance and Raman spectra of these samples were obtained by polarizing microscopy and laser Raman spectroscopy. By means of a comparative study and a graphic method, we discuss in depth whether the TACG samples were affected by intrusion pressure and the attribution of the “Defect” region of these samples.

## 2. Geological Backgrounds

The Huainan Coalfield is located in northern Anhui Province at the southeast corner of the North China Plate (Figure 1a). It is an elongated territory with a mean length of 180 km (W-E) and a width of 15–25 km (N-S). During the Yanshan period, magma intruded extensively into the early Permian coal-bearing strata. Magmatic activities in this period mainly intruded into coal measures through some small and medium-sized dikes and sills, which affected the coal seams in the mine field. Igneous intrusion takes various forms, including intrusion along the coal seam roof, direct cut into the coal seam, and intrusion along the coal seam floor [23]. The Zhuji coal mine is located in the northeast of the Huainan coalfield, with an area of 54.13 km^2^, and its coal reserves are about 950 million tons (Figure 1a). The geological structure of Zhuji coal mine mainly consists of broad shallow dip folds and sporadic faults. The Permian coal-bearing strata in Zhuji coal mine is divided into Shanxi formation, lower Shihezi formation, and upper Shihezi formation. Detailed information on the location and thickness of each coal seam is documented in Figure 1b. Igneous intrusion in Zhuji coal mine is very obvious. Magma intruding into coal-bearing strata ranges from exploration line 6 in the east to exploration line 15 in the west. In addition, the location of igneous intrusion gradually rises from the east No. 3 coal seam to the west No. 8 coal seam. In recent years, the main coal seam mined in Zhuji coal mine was the No. 4 coal seam, located in the east of the mining area. In Figure 1c, the No. 4 coal seam appears obviously affected by igneous intrusion.

## 3. Samples and Experiments

### 3.1. Sampling

Figure 1c shows the position information of four TACG samples at different distances from the intrusion. These samples were collected from the underground working faces of the No. 4 coal seam, and the sampling method used was channeling sampling. After carefully identifying the contact zone between the intrusion and the coal seam, fresh TACG samples were collected, and the gangue was removed.

### 3.2. Coal Quality

In this study, R_max_ (maximum reflectance), R_min_ (minimum reflectance), and R_o_ (mean reflectance) of TACG samples were determined according to GB/T6948-2008. All reflectance values were measured using an optical microscope (Leica DM 6000 M) at a 546 nm wavelength in oil immersion. A resolution of 50/0.85 was used for oil-immersed objects. The test results are listed in Table 1.

In addition, the proximate analysis indexes of coal were measured, including ash content (A_d_, %), volatile matter (V_daf_, %), and moisture (M_ad_, %). These indexes are helpful to analyze and understand the metamorphic degree of coal. As can be seen in Table 1, the ash content of the four coal samples in the study area was between 17.05–19.69%, and the volatile content was between 8.06–8.91%, values typical of low-middle ash coal and low volatile coal. The organic macerals in coal were mainly vitrinite, followed by inertinite and crustite. The inorganic components in coal were mainly clay minerals, followed by carbonate minerals, with scattered sulfides (Figure 2). In addition, it can be seen in Figure 2 that the four coal samples in the study area were affected by magma intrusion, and the coal petrographic components in the coal were often flattened, with a certain directionality, curved cracks, and obvious rheological phenomena (Figure 2c,d).

### 3.3. Field Emission Scanning Electron Microscopy

Pretreatment of the TACG samples and experiments were carried out in the State Key Laboratory of Solid Microstructure Physics, School of Physics, Nanjing University. Sample pretreatment adopted the argon ion polishing sample preparation method. The model of the polishing machine was TIC 3X, and three ion guns were used for plane polishing. Three-beam plane polishing was used, and the surface of the sample was superfinished in a vacuum environment to remove scratches, particles, organic pollutants, and stress damage layers; a flat and crystalline sample surface was obtained for observation of its nanoscale structures. The polishing setting was voltage of 5.0 kV, current of 2.0 mA, and polishing time of 2 h. The polished sample was observed under the ultra-high resolution scanning electron microscope Gemini-SEM 500, and the voltage was set at 2.0 kV. A field emission scanning electron microscope (FESEM) was used to observe the thermal simulation series of coal samples after argon ion polishing treatment, and the pore and mineral distribution characteristics of four metamorphic series of coal samples were analyzed.

### 3.4. Raman Spectroscopy

Raman measurement of the TACG samples were performed with a Raman Spectroscope (Bruker Corporation) equipped with an Nd-YAG laser as an excitation light source. A laser beam of 532 nm was focalized onto the TACG samples to collect the Raman signals in backscattered direction. The laser power was set at 1 mW. The diameter of the laser spot on each analytical sample was 1 μm. Samples with a diameter <56 μm were scanned between 800 cm^−1^ and 2000 cm^−1^, covering the first-order region. The average acquisition time for each spectrum was 45 s.

## 4. Results and Discussion

### 4.1. Organic Matter and Mineral Characteristics

Figure 3 shows a scanning electron microscope photograph of a coal sample. As can be seen, the pores in the coal had various shapes, round and oval, often formed by the dissolution of heated liquid, with burrs and harbor shapes. The pore size of the heat altered pores ranged from <0.1 μm to >10 μm (Figure 3a). It is generally believed that the formation of thermal porosity is related to the generation and escape of volatiles when coal is softened by heat. Radial cracks and sometimes annular cracks are common in the walls and peripheries of some heat-changing pores, and their causes are related to the shrinkage caused by the escape of volatiles in thermoplastic state (Figure 3b,c). We observed globules sprouting around the hot pores and at the bottom of the pores. The particle size and anisotropy of the globules around the pores increased towards the center of the pores. Some pores were filled with hydrothermal minerals such as quartz and calcite, which indicated that the inner walls of these pores had been in contact with the heat-carrying fluid, and the heat energy brought by the heat-carrying fluid caused mesophase globules to sprout from vitrinite (Figure 3d).

Figure 4 shows a scanning electron micrograph of organic matter pores and the mineral composition of the samples in the study area. It can be seen that the structure of the immature intact coal was relatively compact, with few mineral intragranular pores and intergranular pores with organic matter, and only a small number of organic pores were developed, mainly in the shape of ellipses and long strips. Most of the organic pores were of micron size, with a small number of nano-sized pores, and pore sizes fluctuated within the range from 100 nm to 600 nm (Figure 4a). Because coal samples were in a lowly mature state, the pores and fissures in the coal were formed by the compression and dehydration shrinkage of the plant primary pores and the coal by external forces, and micro-nano pores did not developed in the vitrinite. As the distance from the igneous intrusion decreased, the thermal magmatism of the coal increased, and the nano-scale pores in the coal increased, mainly in the form of long strips and ellipses, with pore sizes ranging from 100 to 300 nm. the pore walls were round and smooth and accompanied by a small number of micro-fissures. The micro-fissures were often connected with the nano-scale pores, so that the connectivity of the pores was better (Figure 4b). When the distance was about 10 cm from the igneous intrusion, the pore walls of the fissures in the organic material were rough, which might be caused by the high pressure generated by high-temperature condensation polymerization, which generated coal fissures (Figure 4c).

In addition, the intrusion of the magmatic hydrothermal solution changed the coal in the contact zone with the coal seam into high-rank metamorphic coal and natural coke. The minerals in coal were mainly quartz and kaolinite. At the same time, calcite, pyrite, and other minerals were in contact with the coal seam along the hydrothermal channel of magma, resulting in different mineral compositions in adjacent coals. Therefore, the local distance between samples and magmatic rocks could be inferred from the mineral composition. As can be seen in Figure 4e, the clay minerals in the coal seam were distributed in a strip shape under the influence of temperature and stress caused by magmatic hydrothermal intrusion. The occurrence of ankerite and pyrite in the coal seams close to the magmatic intrusions could be due to late magmatic hydrothermal mineralization (Figure 4f).

### 4.2. Effect of Pressure

Through a study of the reflectance of coal carbonized in the laboratory without excess pressure and under pressure, Chandra [32] proposed that the effect of pressure on TACG could be obtained by measuring the R_ma__x_ and R_min_ of vitrinite. If the reflectance characteristics of TACG are similar to those of residues of coals carbonized in the laboratory without excess pressure, then the pressure effect can be considered negligible or non-existent. If the reflectance characteristics of TACG are different from those of the residues of coals carbonized in the laboratory without excess pressure, the difference indicates the effect of pressure on TACG.

On the basis of the experimental results of Chandra (1965), the R_ma__x_ and R_min_ of TACG were represented in a R_max_ − R_min_ diagram, and the following conclusions were drawn by comparing the R_max_ − R_min_ relationship with that of other TACG (Figure 5). We found that: (1) the R_max_-R_min_ relationship of TACG was similar to that of Wilsontown Main coals (Riddochhill colliery, East Ayrshire, Scotland), and the values of R_max_ and R_min_ for the studied TACG and Wilsontown Main coals were similar to those for coals (85% and 98% carbon content) carbonized in the laboratory without excess pressure. In particular, the R_max_ − R_min_ relationship of TACG was highly consistent with that of 89% carbonized coal. It can be reasonably concluded that the effect of pressure, if any, during thermal metamorphism, on the studied TACG and Wilson town Main coals can be neglected; (2) the results of High Coal Sill coals and Antarctic coals were different from those of coals carbonized in the laboratory without pressure. The R_max_ − R_min_ relationship of the above two coals showed a higher degree of anisotropy than that found in the carbonization test of the full-order series of coals without overpressure in the laboratory. Consequently, these two coals might be subject to overpressure during thermal metamorphism. This conclusion is supported by the observation results of carbonized coal under laboratory pressure.

More generally, there are many forms of igneous intrusion into coal-series strata, which were summarized by Yao and Liu [11]. They concluded that if the magma intrudes along the roof of the coal seam, it releases a large amount of heat to the lower coal seam, causing thermal deterioration of the coal; if the magma is obliquely inserted into the coal seam, brittle deformation of coal occurs due to the existence of a certain “wedge stress”. The Rmax–Rmin relationship of TACG in this study showed that there was no obvious pressure effect, so it can also be suggested that the magma would intrude along the roof of the coal seam, which is consistent with the actual observation results.

### 4.3. Raman Spectra

The first-order Raman spectra of the four graphitized series samples ZJ1, ZJ2, ZJ3, and ZJ4 are presented in Figure 6a. In this paper, three Raman parameters proposed by Tuinstra and Koenig [33] for CM were used: (1) Raman shift frequencies of the *wE*_2g_ spectral peak (G-band) and *wA*_1g_ spectral peak (D-band); the diffraction peaks of each sample were mainly the G and D peaks; (2) FW_G_ of the *E*_2g_ spectral peak and FW_D_ of the *A*_1g_ spectral peak; (3) the ratio of the peak intensity of the *A*_1g_ spectral peak to the peak intensity of the *E*_2g_ spectral peak (*I*_D_*/I*_G_). According to this, after the noise on the experimental spectrogram was removed and using the Fit Multi-peaks function (Lorentzian) in Origin 2017 software, the main Raman parameters of each sample were respectively obtained, as shown in Table 2.

Through spectrum fitting (Figure 6b), it was found that two peaks were fitted in the D-band, namely, the D peak (1320 cm^−1^) caused by C–C bond vibration between aromatic ring and aromatic compound with not less than six rings, and the D_s_ peak (1360 cm^−1^) [29], which was the sub-peak of the D peak and mainly derived from the stretching vibration of amorphous carbon structure. Similarly, two peaks were fitted in the G-band, namely, the G peak (1580 cm^−1^), caused by the aromatic ring stretching vibration and the alkene C=C bond vibration [28], and the G_s_ peak (1600 cm^−1^), which was the sub-peak of the G peak [29]. In the following analysis, for convenience of expression, the *A*_1g_ peak (D-band) corresponds to the fitted D peak, and the *E*_2g_ peak (G-band) corresponds to the fitted G peak [30]. The above-mentioned shift frequencies, FWHM, and peak intensity ratios were all obtained from the fitted D- and G-peak.

A group of coal-based graphite samples with different graphitization degrees are shown in Figure 6a. As mentioned earlier [27,34,35], the 1580 cm^−1^ spectral peak of this spectrogram was a two-dimensional lattice stretching vibration peak, which is the natural spectral peak (G peak) of graphite. Obviously, the G peak is relatively stable, its intensity increases with the improvement of the coal-based graphite lattice, and its peak shape is sharp. This was clearly shown by the spectral peaks of the samples shown in Figure 6a. Compared with other samples, the ZJ4 sample had a low graphitization degree, accompanied by low G peak intensity and narrow G peak width (FW_G_ = 29 cm^−1^), which means that its lattice integrity was not high. With the increase of the graphitization degree, the G peak of the ZJ-3, ZJ2, and ZJ1 sequence samples became higher and sharper, and the corresponding FW_G_ decreased from 27 cm^−1^ to 20 cm^−1^.

According to the calculation and analysis by [33], the spectrum peak near 1320 cm^−1^ was the *A*_1g_ peak, which was caused by structural defects. As can be seen in Figure 6b, the intensity of the D peak near 1320 cm^−1^ gradually weakened with the increase of the graphitization degree and gradually became a secondary spectral peak compared with the G peak. Therefore, the decrease of the spectral peak near 1320 cm^−1^ indicated that the structural defects of the peak steadily disappeared and were replaced by the integrated lattice region during the graphitization process. Therefore, the structural defects reflected by this spectral peak had the characteristics of primary structural defects.

### 4.4. Relationship between L_a_ and FW_G_ and FW_D_

Tuinstra and Koenig [33] considered that the *A*_1g_ spectrum peak was the carbon layer diameter effect of graphite crystallites. Due to the small crystallite size, the *A*_1g_ vibration turned into Raman activity, and the 1320 cm^−1^ spectrum peak (D peak) was generated. From this, the intensity of the D peak is proportional to the diameter L_a_ of the crystallite carbon layer
(1)$$L_{a}=44/\left(I_{D}⁄I_{G}\right)$$
where the unit of L_a_ is Å (10^−1^ nm), and I_D_ and I_G_ are the intensities of D- and G-peak, respectively. According to this formula, the Raman spectrum data of each sample listed in Table 2 were used to obtain the corresponding L_a_ values, which are listed in the right end of Table 2.

The L_a_-FW_D_ and L_a_-FW_G_ diagrams are shown in Figure 6a,b. In Figure 6, it can be observed that the graphitization process increased with the increase of L_a_, but FW_G_ and FW_D_ decreased gradually, as shown in the graphitization process curve in (Figure 7).

Vibration analysis showed [36,37] that the factor group analysis of the integrated graphite lattice space group is the basis for understanding the Raman spectra of graphitized CM. Graphite cells have four carbon atoms, which has nine vibration modes, described as:(2)$$2B_{1g}\left(-\right)+2E_{2g}\left(R\right)+A_{2g}\left(IR\right)+E_{1u}\left(IR\right)$$
where (−) is infrared and Raman inactivity; (*R*) is Raman activity; (*IR*) is infrared activity.

For other graphite materials that are not single crystals, only one layer of carbon atoms needs to be analyzed, because the vibrational coupling between the graphite carbon layers is very weak and can be omitted. Therefore, graphite is considered a two-dimensional lattice with two carbon atoms in the cell, and its two vibrational modes can be described as:(3)$$B_{1}\left(-\right)+E_{2g}\left(R\right)$$

Therefore, the two-dimensional lattice graphite has only one *E*_2*g*_ vibration mode. In this mode, the reciprocal points within a layer of carbon atoms and on its boundaries are possible vibrational phonons. For ideal graphite lattice, the vibration phonon symmetry is high, and its vibration frequency can be cancelled or strengthened. Therefore, the phonon vibration on the graphite carbon layer cannot cause a change in polarization, exhibiting non-Raman activity (only the G peak vibration). That is, as the diameter of the carbon layer (L_a_) increases gradually, the higher the symmetry of the vibration phonons on the carbon layer, the lower the polarization caused by the vibration, the higher the intensity of the G peak, and the lower the FW_G_ value. This is consistent with Figure 6a. The regression equation between L_a_ and FW_G_ of coal-based graphite sequence samples was obtained in this study:(4)$$L_{a}=-0.09948{\mathrm{FW}}_{G}+35.02849\;(\mathrm{Correlation}\;\mathrm{coefficient}\;r=0.84)$$

Similarly, for finite-size graphite crystallites, the *A*_1*g*_ mode will change to Raman activity, and the D peak vibration caused by the boundary defects of graphite carbon layer will be strengthened. That is, as the diameter of the carbon layer (L_a_) increases gradually, the mode *A*_1*g*_ will change to non-Raman activity, the vibration effect of D peak will decrease, and its intensity and FW_D_ value will also decrease. This is consistent with Figure 6b. The regression equation between L_a_ and FW_D_ of the coal-based graphite sequence samples was obtained in this study:(5)$$L_{a}=-0.13135{\mathrm{FW}}_{D}+48.99568\;(\mathrm{Correlation}\;\mathrm{coefficient}\;r=0.95).$$

In accordance with the Raman spectrum characteristics of coal-based graphite, the D band of graphite can be divided into two types of structural defect bands with different properties [33]. One is the original structure defect band; the other is the secondary structure defect band. According to Nakamizo et al. [37], the latter is usually caused by shear pressure or grinding. On this basis, combined with Figure 6, it can be concluded that the D peak of the studied TACG samples could be caused by its original structural defects.

## 5. Conclusions

The main conclusions of this study are as follows: (1) affected by the temperature and stress of igneous intrusion, the clay minerals in the coal seams were distributed in strips. The occurrence of ankerite and pyrite in the coal seams near the magmatic intrusions could be due to late magmatic hydrothermal mineralization; (2) The R_max_–R_min_ correlation confirmed that four TACG were not subjected to obvious magmatic shear pressure; (3) with the increase of the graphitization process, the positions of the D- and G-peak shifted to a lower wavenumber. The intensity of the G peak increased, and the spectral pattern was sharp, while the intensity of the D peak decreased, and the spectral pattern was low. The FWHM values of the D- and G-peak decreased; (4) the FWHM of the D- and G-peak showed a significant positive correlation with L_a_, indicating that with the intensification of the graphitization process, the degree of the lattice defects in coal-based graphite decreased, and the crystal form tended to be integrated.

## Figures and Tables

**Figure 1 molecules-27-03810-f001:**
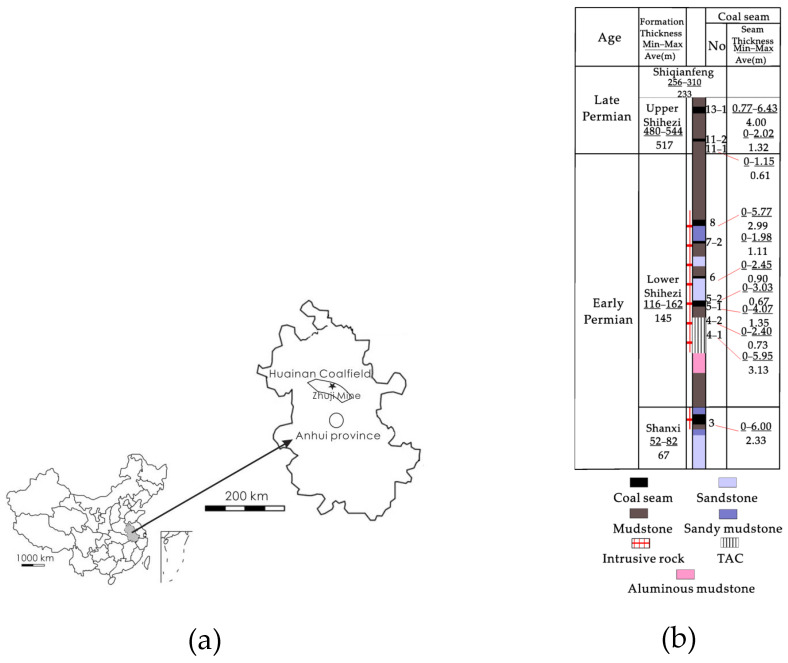
Geographical location (**a**), stratigraphic column (**b**), and sampling (**c**) of the study area.

**Figure 2 molecules-27-03810-f002:**
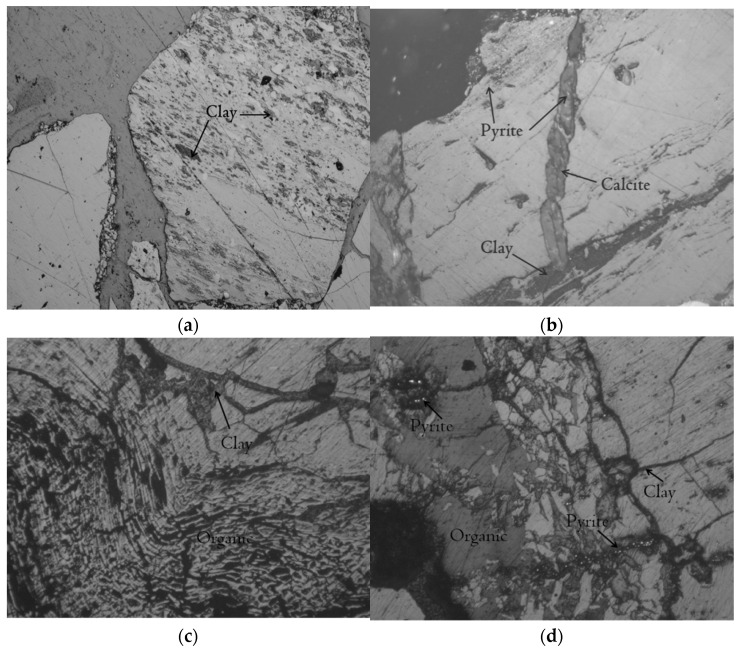
Microscopic coal petrography. (**a**) ZJ4 sample; (**b**) ZJ3 sample; (**c**) ZJ2 sample; (**d**) ZJ1 sample.

**Figure 3 molecules-27-03810-f003:**
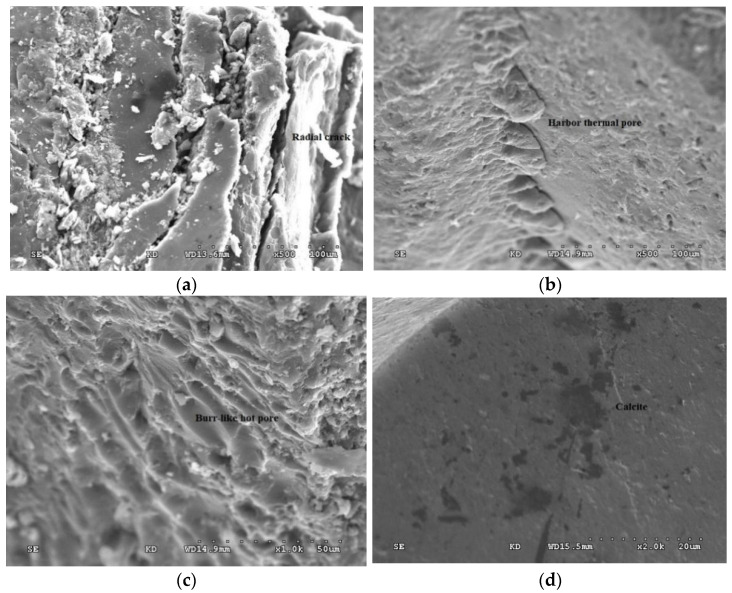
SEM of coal samples. (**a**) burrs shapes and radial crack; (**b**) hot altered hole; (**c**) burr-like hot pore; (**d**) calcite fills hydrothermal pore.

**Figure 4 molecules-27-03810-f004:**
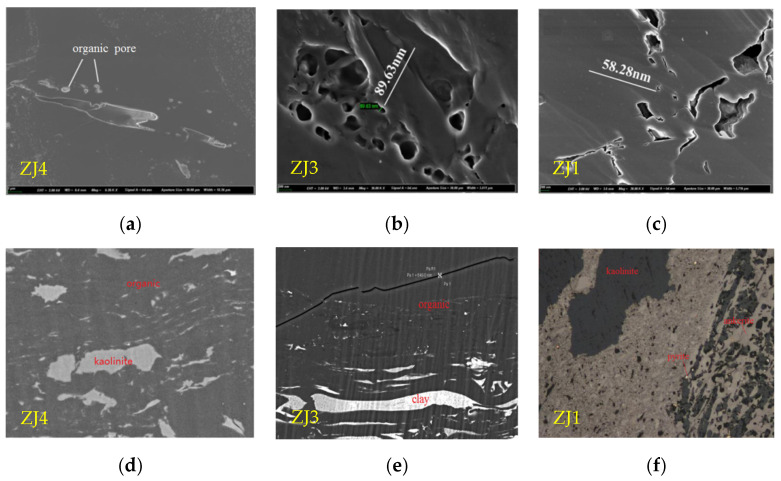
Changes of pore morphology and mineral composition in coal at different distances from the intrusive rocks. (**a**) micro-nanometer organic pore; (**b**) round and smooth pores; (**c**) high temperature condensation pore; (**d**) hydrothermal mineral; (**e**) banded clay minerals; (**f**) ankerite and pyrite.

**Figure 5 molecules-27-03810-f005:**
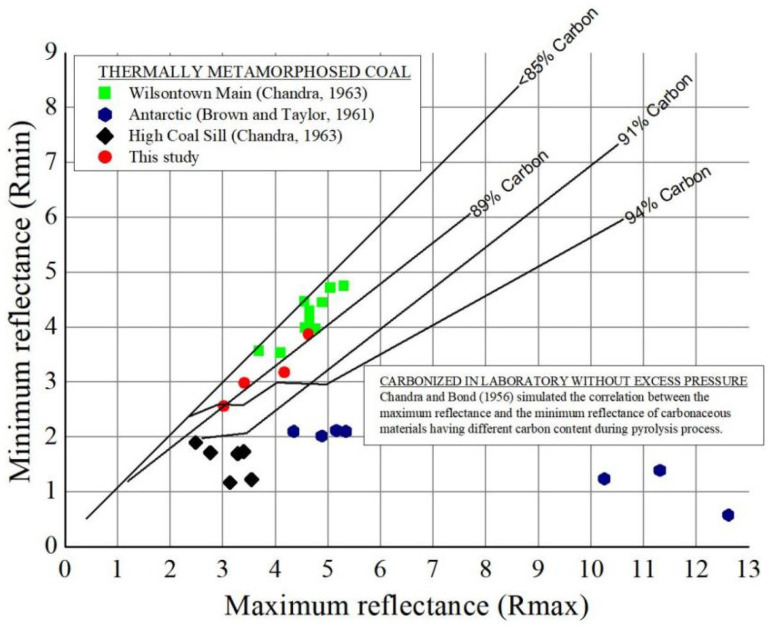
Relationship between R_max_ (%) and R_min_ (%).

**Figure 6 molecules-27-03810-f006:**
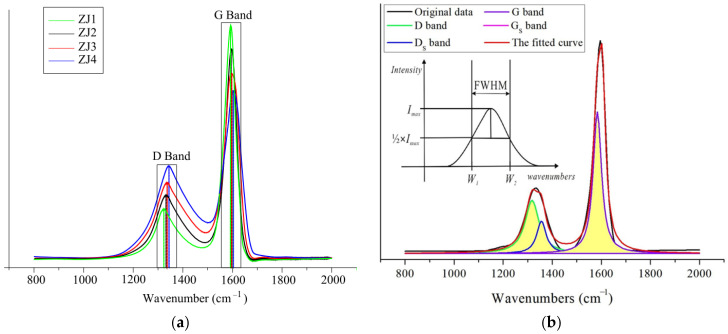
Raman spectra of TACG samples and Raman spectrum fitting of the ZJ2 sample. (**a**) Raman spectra of four graphitized series samples; (**b**) Raman spectrum fitting results of Z3 sample and the corresponding FWHM parameter settlement principle.

**Figure 7 molecules-27-03810-f007:**
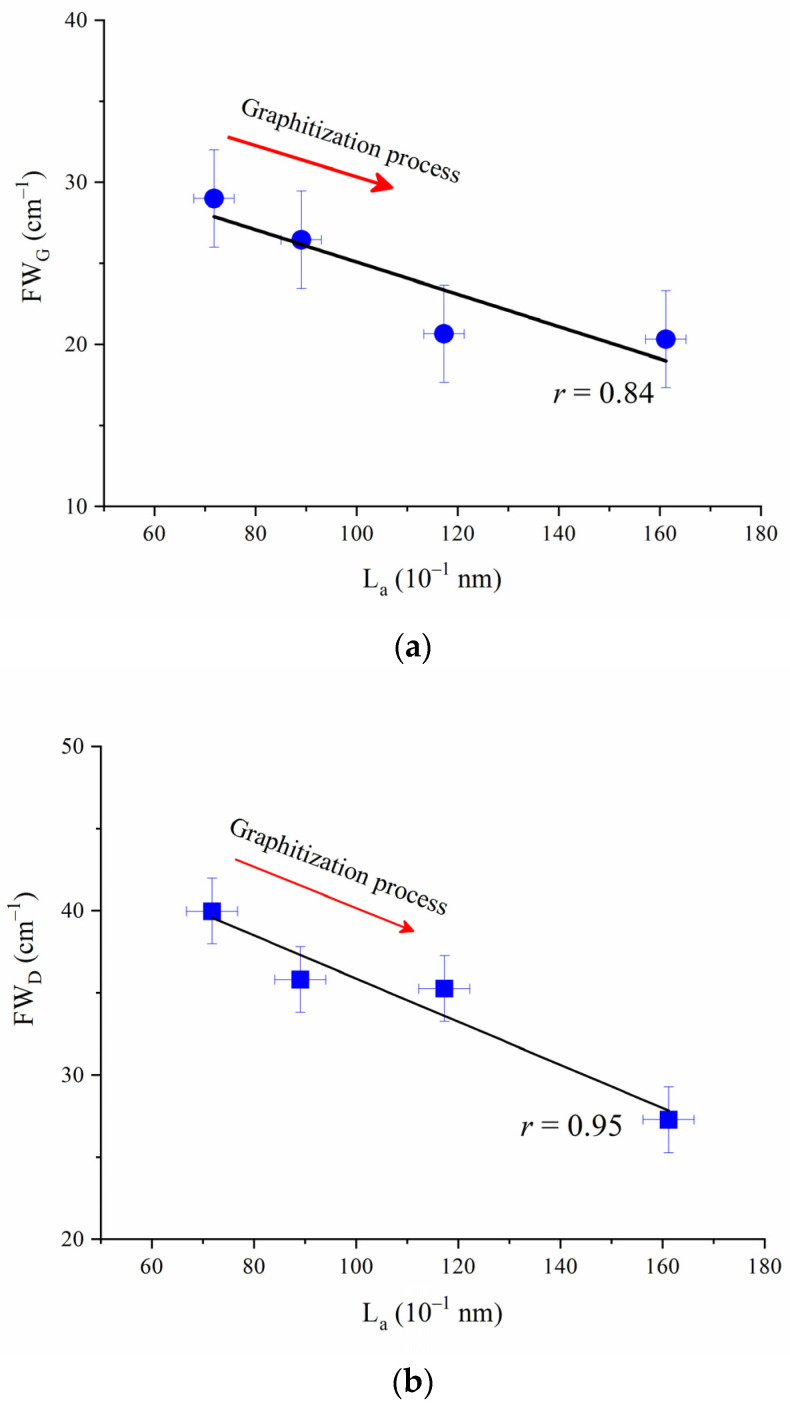
Correlation between L_a_ and FW_G_ (a) and between L_a_ and FW_G_ (**b**).

**Table 1 molecules-27-03810-t001:** Basic data of the coal-based graphite series samples.

Sample	D (m)	R_min_ (%)	R_o_ (%)	R_max_ (%)	Proximate Analysis			Organic			Inorganic		
					Ad (%)	Vdaf (%)	Mad (%)	Vitrinite (%)	Inertinite (%)	Liptinite (%)	Clay (%)	Sulphide (%)	Carbonate (%)
ZJ1	0.1	3.87	3.91	4.63	17.46	8.86	0.64	72.34	20.26	4.11	1.36	0.50	1.43
ZJ2	0.2	3.18	3.55	4.17	19.69	8.91	0.67	47.62	35.88	11.17	2.50	0.70	2.13
ZJ3	0.4	2.98	3.12	3.41	17.05	8.74	0.54	56.35	31.02	7.94	3.30	0.46	0.93
ZJ4	0.7	2.56	2.78	3.02	19.66	8.06	0.62	62.70	23.41	5.83	3.37	0.91	3.79

D: distance from the sill boundary; d: dry basis; daf: dry-ash-free basis; ad: air-dry basis.

**Table 2 molecules-27-03810-t002:** Raman spectroscopic parameters of the coal-based graphite series samples.

Sample	Band	Position (D- and G-Peak)	I_D_/I_G_	FWHM (D- and G-Peak)	L_a_ (Å)
ZJ1	D peak	1309	0.273	27.272	161.2
	G peak	1580		20.323	
ZJ2	D peak	1317	0.375	35.259	117.3
	G peak	1583		20.653	
ZJ3	D peak	1322	0.494	35.805	89.1
	G peak	1584		26.459	
ZJ4	D peak	1329	0.613	39.963	71.8
	G peak	1591		28.992	

## Data Availability

Not applicable.

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
