# Peer review of "Mineral Composition and Graphitization Structure Characteristics of Contact Thermally Altered Coal"

_molecules, 2022, doi:10.3390/molecules27123810_

Round 1

Reviewer 1 Report

The content presented in the paper is very interesting. As well pointed by the author the understanding of parameters impacting graphitization process of coal affected by contact metamorphism poses scientific and practical significance.  The core part of the paper is a description and discussion of FESEM tests with Raman measurement of TACG samples. The testing procedure is presented clearly. The paper starts with an interesting introduction part where authors provide a robust state of the art. Authors provide also good reference list. Additionally, the language quality in the paper does not raise any objections.

The drawback of the publication is quality of figures that might be more readable. Also the determination of relationship in chapter 4.4 only based of four measurement is questionable. It requires at least additional comment. Moreover the conclusions lapidary it could be enriched by one or two sentences on meaning of the research and further work to be done.

To summarize the paper is good for publication after minor revision.

Author Response

Thank you for your comments. We will further improve the pictures in the text and make them easier to understand. Affected by the support of coal mining sampling face, there are few exposed coal seams, and only exposed coal seams are found in the intrusion part of magmatic rocks, with the exposed length of about 1m. Therefore, only four samples from different positions of magmatic rocks were collected this time. We have adopted your opinion and made supplementary explanations in the sampling chapter of the manuscript. Some conclusions have been improved according to the comments of reviewers.

Reviewer 2 Report

This MS is of geo-petrological interest. Nevertheless, an improvement should be made, at least, in Section 4.2 "Effects of pressure", where a reference is missing, in the text, to Figure 3.    Further, the lower part of this Figure needs deeper comments.

Author Response

Thank you for your comments. Due to our carelessness, Figure 3 has been marked, and the content of Figure 3 is further discussed.

Reviewer 3 Report

The manuscript entitled " Mineral composition and graphitization structure characteristics of contact thermally altered coal” by Huogen Luo et al. describes the structural changes of TACG at different distances from an intrusive body within the Zhuji coal mine of Huainan Coalfield in North China.

Although the manuscript is interesting,  the total number of samples analysed in this study is to small - total of 4. I think that collecting a set of four aligned samples with a 70 cm apart is not enough to evaluated the process of an igneous intrusion of 5 km of extension within a large kilometric coalfield. With this samples collection we can not place much confidence in the significance of the conclusions. In my opinion the authors should collect and analyze a larger number of samples and not only in a 70 cm apart to the igneous contact, and re-write the paper with the benefit of a much better data set.

Additionally, the characteristics of unaltered coal should be presented.

The authors measured vitrinite reflectance, however optical microscopy pictures of the thermal effect in this particles are not shown.

The authors present in table 2 the Raman spectroscopic parameters D and G bands. Raman parameters obtained (e.g. D position, G position, FWHM) should be given in integer (no digit after the dot !): due to the resolution of the equipment used the precision is not so good !!! 

Finally, in Fig. 4b the authors present an example of Raman spectrum fitting of ZJ2 sample using Ds and Gs bands. The reason for using this deconvolution procedure on this “graphitic material” is not clear.

Author Response

  1. Thank you for your valuable opinion. As for the number of samples, due to the influence of the support of coal mining sampling face, there are few exposed coal seams, and only exposed coal seams are found at the intrusion part of magmatic rocks, with the exposed length of about 1m. Therefore, only four samples from different positions of magmatic rocks are collected this time. Because, in this paper, our idea is to focus on one point, and focus on the research and understanding of the influence of magma intrusion on the graphitization degree of coal. Your advice is very good. In the following scientific research, we will constantly improve and perfect the experimental scheme, and strive to achieve more valuable results.
  2. Thank you for your valuable comments. We have supplemented and improved the characteristics of raw coal.
  3. Thank you for your valuable comments. We have supplemented the relevant optical micrographs and explained the thermal effect.
  4. Thank you for your valuable opinion. The equipment used in our Raman spectroscopy experiment is a Raman spectrometer (Bruker Corporation) equipped with Nd-YAG laser as the excitation light source. The accuracy of the equipment can meet the requirements of experimental test and analysis. Maybe we didn't give the relevant parameter values of the peak according to the integer in the data processing process, and this part has been modified according to your opinion.
  5. Thank you for your valuable comments. We have explained the relevant reasons.

Reviewer 4 Report

The issues presented in this article have been previously studied and published by other authors (regarding different locations). Therefore, the added value of the manuscript is very limited. This is just another example of how the heat and pressure of magmatic intrusion affects the microstructure of coal. 

Major remarks:

  1. The number of 4 samples used in the study is too small to form firm conclusions.
  2. The number of spectra obtained from each sample is not given. This is an important deficiency as it is not possible to assess if the statistical evaluation of the results is possible.
  3. Line 194 - the Authors write that "a large number of pores, but also fissures are developed". It is surely not possible to assess based on the one SEM microphotograph.
  4. Line 202-203 - the Authors write that "the above-mentioned minerals such as illite, calcite, pyrite, ankerite and andalusite penetrate into the coal seam". These minerals were not mentioned anywhere "above". What is more most of them cannot penetrate into the seam themselves. These are hydrothermal solutions that migrate. Minerals precipitate from them.
  5. Table 2 - the La value is calculated from the Tuinstra and Koenig equation. To assess graphitization process, it would be much better to determine La from HRTEM observation. G band parameters for the ZJ4 sample are not given in the table.
  6. The reference to the Gs band (at 1600 cm-1) may not be adequate. The paper cited (Menella et al., 1995) describes the Raman spectra of CM excited at 1064 nm. The Authors carried out their measurements using the 532 nm Nd-YAG laser. The laser wavelength affects the spectrum. Several works exist which attribute the 1600-1620 cm-1 G2 band to the E2g vibrations of the graphite ring (together with the G band at 1580 cm-1) but not to the C=O bonds.
  7. The results obtained with the Raman and La methods are not discussed at all with reference to previous work on thermally altered coals, which is a serious shortcoming.

Minor remarks:

  1. The standard (ISO?) used in the reflectance measurements is not given. It is difficult to compare the results with those obtained by other authors.
  2. Some parts of text and tables are highlighted. Why?
  3. There are mistakes with the units in line 159. 
  4. Sentence in lines 192-197 is very long and not entirely clear.
  5. Lines 259-260 reference to the Ds band needed.
  6. Line 298 - there are mistakes in the references to the figures.
  7. Line 300 - FWD decreases, not increases.

Author Response

Reply(Mark in purple font in the article.):

Major remarks

1.Regarding the number of samples, we didn't make it clear in the previous reply. I would like to express my sincere apologies to you here. Coal samples near magma contact zone are collected in the area exposed by underground working face. The original number of samples collected was about 8, but due to the complicated and difficult underground conditions, when we checked and analyzed these 8 samples, we found that 4 samples were seriously ashed and were actually coal gangue, which had no great reference value for this study. The other four samples are metamorphic coal, and then the follow-up study was made.
The coal mine where we collected the samples belongs to the mine with high gas outburst. As you know, there are some safety problems that will be faced when the working face is advanced and sampling in the next well, such as gas outburst. Therefore, there are not many samples in this study, but some methods are mainly used to put forward the mineralogical and spectroscopic properties of graphitized coal, which lays the foundation for future research.
The "future research" in the previous reply mainly indicates that we will collect more samples for analysis in other working faces.

2.We have improved the relevant content presentation.

3.We have corrected the relevant statement.

4.Thank you for your comments. We have corrected the relevant statements.

5.We have improved the relevant content presentation.

6.We have improved the relevant content presentation.

7.We have improved the relevant content presentation.

Minor remarks

1.The measurement of vitrinite reflectance of coal in this work is based on GB/T6948-2008.

2.It has been corrected. This is the key point in the process of revision.

3.It has been corrected.

4.We have perfected the expression of the words here.

5.We have added relevant references (Mennella et al., 1995). It is consistent with the citation of Gs.

6.We carefully examined the format of references related to the drawings (Hiura et al., 1993; Cuesta et al., 1994; Sonibare et al., 2010), and the corresponding amendments were made.

7.It has been corrected.

Reviewer 5 Report

The manuscript of Huogen Luo  and co-authors is devoted to the study of structural changes in thermally altered coal-based graphite (TACG). The authors have established graphitization factors, features of Raman spectra of TACGs.

The manuscript is well-structured, easy to follow and is supplied by good illustrations. The presented data are of sufficient quality and the discussion is well developed and focused. This manuscript seems suitable for publication in the special issue “Molecular Structure of Minerals”.

As a comment, it is necessary to check the correctness of the specified units of measurement of the diameter of the laser spot and the size of the sample (line 159).

Author Response

Reply:(Mark the article in red font.)

The relevant content has been checked. The laser spot diameter is 1 μ m.. Sample diameter < 56 μ m.

Round 2

Reviewer 3 Report

The authors add several things to the manuscript such as Ad (%) Vdaf (%) Mad (%) without any explanation of their meaning but still very incomplete since no maceral content is referred. They have added SEM pictures instead of optical microscopy pictures as asked.

Within the Raman parameters the authors correct the precision for position but not for the FWHM.

The reason for using the Ds and Gs bands on deconvolution procedure of the “graphitic material” (Fig.4b) is still not clear.

Finally, the authors explained that there are few exposed coal seams, and only four samples were collected this time.  However, as I stated before 4 samples are not enough to evaluated the process of an igneous intrusion of 5 km of extension within a large kilometric coalfield. The authors also explain that in the following scientific research, they will constantly improve and perfect the experimental scheme, and strive to achieve more valuable results. I think this is a wrong way to do research by saying that next time it will be done properly.

Author Response

Reply(Mark the article in green font.):

1.We added the information of proximate analysis in this article. The optical image of coal maceral is supplemented.

2.We have corrected the accuracy of FWHM parameters.

3.In fig. 5(a), we can observe that the Raman spectrum of coal samples is mainly divided into two bands, namely D band and G band. However, there are major peaks and minor peaks in D-Band and G-Band that we need to analyze. In order to obtain these information, Raman spectra need to be fitted by peak separation. In fig. 5(b), we obtained the main peak (D peak and G peak) and the secondary peak (Ds peak and Gs peak) in the D and G bands, respectively, but in the later calculation, we used the peak position and FWHM information of the former. At the same time, the strength of Ds and Gs also reflects the degree of coal graphitization. We updated the peak information in Table 2.

3.Regarding the number of samples, we didn't make it clear in our previous reply. I would like to express my sincere apologies to you here. Coal samples near magma contact zone are collected in the area exposed by underground working face. The original number of samples collected was about 8, but due to the complicated and difficult underground conditions, when we checked and analyzed these 8 samples, we found that 4 samples were seriously ashed and were actually coal gangue, which had no great reference value for this study. The other four samples are metamorphic coal, and then the follow-up study was made.
The coal mine where we collected the samples belongs to the mine with high gas outburst. As you know, there are some safety problems that will be faced when the working face is advanced and sampling in the next well, such as gas outburst. Therefore, there are not many samples in this study, but some methods are mainly used to put forward the mineralogical and spectroscopic properties of graphitized coal, which lays the foundation for future research.
The "future research" in the previous reply mainly indicates that we will collect more samples for analysis in other working faces.

Reviewer 4 Report

The Authors presented much improved version of their manuscript. The comments included in the review have been teken into account, and several changes have been made. The Authors' response is clear and satisfying. Therefore, the overall quality of the paper is much higher, now.

I suggest to accept the manuscript in present form.

Round 3

Reviewer 3 Report

The authors of the manuscript revised it according to the comments and I think it can be published in the journal.